# Fine Mapping and Functional Verification of the *Brdt1* Gene Controlling Determinate Inflorescence in *Brassica rapa* L.

Cuiping Chen [1,†], Xuebing Zhu [1,2,†], Zhi Zhao [1,2], Dezhi Du [1,2] and Kaixiang Li [1,2,*]

1   Academy of Agricultural and Forestry Sciences of Qinghai University, Xining 810016, China; chencuiyang@126.com (C.C.); 15297190798@163.com (X.Z.); zhaozhi918@sohu.com (Z.Z.); qhurape@126.com (D.D.)
2   Key Laboratory of Spring Rape Genetic Improvement of Qinghai Province, Rapeseed Research and Development Center of Qinghai Province, Xining 810016, China
*   Correspondence: 18997174190@163.com; Tel.: +86-0971-5366-520
†   These authors contributed equally to this work.

**Abstract:** *Brassica rapa*, a major oilseed crop in high-altitude areas, is well known for its indeterminate inflorescences. However, this experiment revealed an intriguing anomaly within the plot: a variant displaying a determinate growth habit (520). Determinate inflorescences have been recognized for their role in the genetic enhancement of crops. In this study, a genetic analysis in a determinate genotype (520) and an indeterminate genotype (515) revealed that two independently inherited recessive genes (*Brdt1* and *Brdt2*) are responsible for the determinate trait. BSA-seq and SSR markers were employed to successfully locate the *Brdt1* gene, which is localized within an approximate region 72.7 kb between 15,712.9 kb and 15,785.6 kb on A10. A BLAST analysis of these candidate intervals revealed that *Bra009508* (*BraA10.TFL1*) shares homology with the *A. thaliana TFL1* gene. Then, *BraA10.TFL1* (gene from the indeterminate phenotype) and *BraA10.tfl1* (gene from the determinate phenotype) were cloned and sequenced, and the results indicated that the open reading frame of the alleles comprises 537 bp. Using qRT-PCR, it was determined that *BraA10.TFL1* expression levels in shoot apexes were significantly higher in NIL-520 compared to 520. To verify the function of *BraA10.TFL1*, the gene was introduced into the determinate *A. thaliana tfl1* mutant, resulting in the restoration of indeterminate traits. These findings demonstrate that *BraA10.tfl1* is a gene that controls the determinate inflorescence trait. Overall, the results of this study provide a theoretical foundation for the further investigation of determinate inflorescence.

**Keywords:** *Brassica rapa*; *Brdt1*; determinate inflorescence; morphological observation; paraffin sectioning; mapping; quantitative real-time PCR; transformation

## 1. Introduction

Oil crops belonging to the genus Brassica hold significant importance in global agriculture. This genus comprises a diverse range of plant species, including the species *Brassica rapa* L., *Brassica napus* L., and *Brassica juncea* L. [1,2]. *Brassica rapa* is a diploid species (2n = 20, AA) and has a worldwide distribution [3]. It includes vegetable crops and oil seed crops [4]. Oil seed crops have excellent characteristics, such as barren, drought, and cold resistance [5,6]. Currently, the rapeseed cultivation area in China is approximately seven million hectares, with *B. rapa* accounting for about 15% of this area [7]. *Brassica rapa* plays an irreplaceable role in the provinces of the Yangtze River basin and the Northwest Plateau due to its short growth period, particularly in the rotation of rice and rapeseed. All *Brassica* crops possess an indeterminate growth habit, which is influenced by competition for resources within the plant canopy, both within and between plants [8]. This competition often leads to incomplete seed filling, immature pods, and sterility at the tip of the plant at maturity [9]. Additionally, certain varieties with indeterminate inflorescence growth habits have drawbacks that negatively impact yield, such as taller plants, an increased

vulnerability to lodging, longer growth periods, and inconsistent ripening periods [10]. Transforming plants from indeterminate to determinate inflorescence offers a new approach for breeders. Thus, it is necessary to research the molecular mechanisms underlying the determinate inflorescence traits in *B. rapa* to enhance rapeseed genetics.

There are many research reports on determinate inflorescence [8,11–13]. Kaur and Banga [8] identified the determinate gene *Sdt1*, which was responsible for regulating determinate inflorescence on the B5 chromosome of *Brassica juncea*. Li et al. [9,14] discovered a double haploid (DH) line 4769 that exhibited a determinate inflorescence trait. Further investigation revealed that this trait was controlled by two independently inherited recessive genes (*Bnsdt1* and *Bnsdt2*). Wan et al. [15] employed BSA-Seq technology to identify two QTL loci associated with determinate inflorescence in a mutant of *B. napus*. These loci were found on the C02 and C06 chromosomes, respectively. Chen et al. [10] discovered that the determinate inflorescence natural mutant 6138 material of *B. napus* is controlled by a single recessive gene, *BnDM1*. Furthermore, they successfully identified the location of the gene on the C02 chromosome.

The *TERMINAL FLOWER 1* (*TFL1*) gene and its homolog play an important role in determinate inflorescence. The genetic mechanism of the *TFL1* gene in *A. thaliana* has been extensively studied [16–18]. The *TFL1* gene is predominantly expressed in the central region of the apical meristem of *Arabidopsis* [19]. Its interaction with genes, such as *LEAFY* (*LFY*), *APETALA1* (*AP1*), and *FLOWERING LOCUS T* (*FT*), influences the development of the stem tip of *A. thaliana*, leading to the formation of distinct inflorescences [20,21]. *TFL1*, being the main inhibitor of flower development, interferes with the development of flowers by binding to the transcription factor *FLOWERING LOCUS D* (*FD*) and inhibiting *FT*. This inhibition is crucial in regulating the expression of downstream flowering integrators *AP1* and *LFY* [22].

Additionally, there are many reports about *TFL1* in other plants. In rice, the *TFL1* homologous gene *RCN* initiates its regulatory pathway by binding to *14-3-3* proteins, followed by its interaction with *FD* to regulate flower recognition genes [23]. However, *CsTFL1* in cucumbers does not directly interact with *CsFD*, *CsFDP*, or *Cs14-3-3*. Instead, it interacts with *CsNOT2a*, binding *CsFD* or *CsFDP* and affecting the transcription of *CsAP1* and *CsLFY* [24]. Therefore, it can be concluded that the regulatory pathway of *TFL1* in tall plants is a relatively complex process.

Although there has been extensive research on determinate inflorescence in other plants [25–27], the genetic patterns and molecular basis of determinate inflorescence remain unclear in *B. rapa*. In order to investigate the genetic inheritance patterns controlling determinate inflorescence in *B. rapa*, this study focused on the determinate inflorescence mutant 520 of *B. rapa*. Two genes (*Brdt1* and *Brdt2*) associated with determinate inflorescence were identified and further subjected to the fine mapping and functional verification of *Brdt1*. These findings enhance our theoretical understanding of determinate inflorescence and lay the groundwork for future investigations.

## 2. Materials and Methods

### 2.1. Plant Material and Population Construction

Lines 520 and 515 of *B. rapa* were used as materials in the present study. The inflorescence of 520 was determinate (a natural mutant) (Figure 1a), while the inflorescence of 515 was indeterminate (Figure 1b). At maturity, they will exhibit different morphological structures (Figure 1c,d). Initially, $F_1$ generation was obtained through a hybrid cross between lines 515 and 520, completed in 2018. Subsequently, they were selfed to produce the $F_2$ generation, and the $F_1$ generation was backcrossed with the recessive parent line 520 to generate the $BC_1F_1$ generation (2019). The $F_2$ and $BC_1F_1$ populations were used for the genetic analysis of inflorescence traits. The inflorescence traits were investigated during the flowering period in 2020.

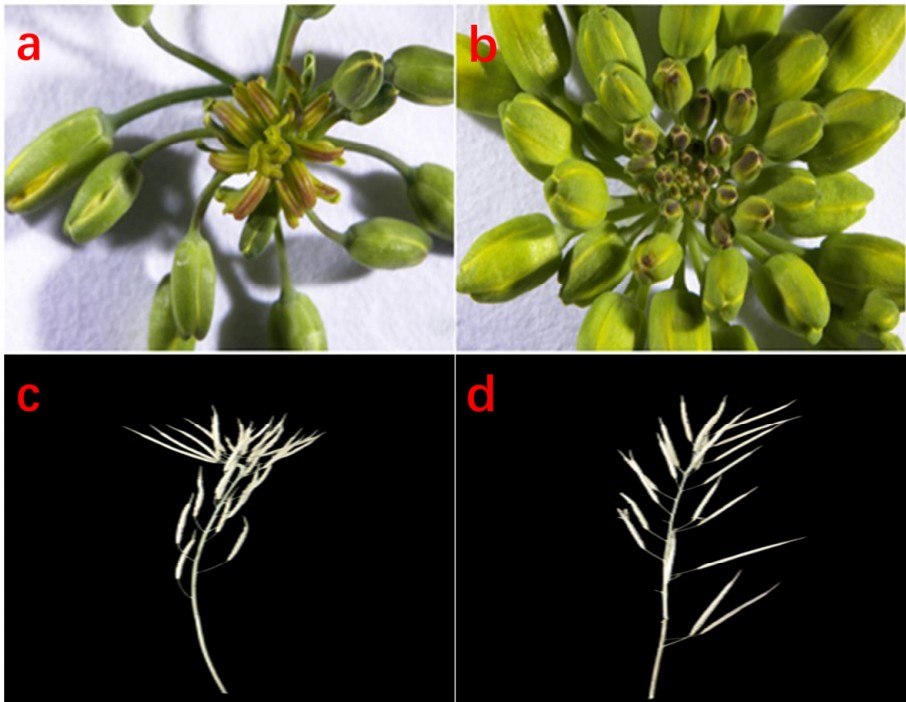

**Figure 1.** Two IM development phenotypes in *B. rapa*. (**a**) The determinate phenotype (520). (**b**) The indeterminate phenotype (515). (**c**) The line 520 phenotype at maturity. (**d**) The line 515 phenotype at maturity.

The *Brdt1* gene was identified using the $BC_1F_1$ population. From this population, 12 indeterminate individuals were selected for backcrossing with line 520. The resulting $BC_2F_1$ isolate line was then planted at the Xining Experimental Base in Qinghai Province, China, in 2020. Subsequently, the inflorescence traits of each line were investigated and it was found that the indeterminate to determinate separation ratio was 1:1 in 9 out of 12 lines. The dominant *BrDT1* gene was identified in an isolated line, with an indeterminate to determinate ratio of 1:1. To establish the $BC_3F_1$ population, 10 indeterminate individuals were selected for backcrossing with individuals from line 520, resulting in a population of 847 individuals (2021). From this $BC_3F_1$ population, 18 indeterminate individuals were chosen and backcrossed with individuals from line 520 to create the $BC_4F_1$ population, which consisted of 1267 individuals (2022). This $BC_4F_1$ population was used for the fine mapping of the *Brdt1* gene (Figure S1). Additionally, near-isogenic lines were constructed for gene expression analysis, including line 520 (determinate) and NIL-520 (indeterminate).

*2.2. Morphological Observation and Paraffin Sectioning*

The inflorescence variations were observed under the fluorescent microscope (4× magnification), based on stem apex meristem development in accordance to the method described by Kobayashi et al. [28]. In this study, a stereoscopic fluorescence microscope (Nikon SMZ25, Tokyo, Japan) was used to observe the inflorescence SAM in homozygous plants at different stages, including the 2-, 4-, 6-, and 8-leaf stages. The aim was to investigate the differences in apical SAM. The methods for paraffin sectioning plant SAMs were followed as described in previous studies [29]. The samples were treated with 70% formalin-acetic acid-alcohol (FAA), followed by Safranin O staining, rinsing, dehydration, clearing, infiltration, embedding, slicing, sealing, and examination using a Nikon microscope (Nikon, Tokyo, Japan).

### 2.3. DNA Extraction and BSA-Seq

DNA was extracted individually from fresh leaves using the cetyltrimethylammonium bromide (CTAB) method [30]. To map the *Brdt1* gene, 20 plants with indeterminate inflorescence and 20 plants with determinate inflorescence were selected from the $BC_3F_1$ population. Separate indeterminate and determinate bulks were created. The two parents and bulks were subjected to bulked segregant analysis (BSA) at Novogene Biological Company in Beijing, China. The Illumina HiSeq TM PE150 sequencing method was employed. Subsequently, the Burrows–Wheeler alignment (BWA) tool aligned the whole-genome sequencing (WGS) reads to the reference genome of 'chiifu' v1.5 (BRAD (http://brassicadb.org/brad/)). Single-nucleotide polymorphisms (SNPs) were detected using the Haplotype Caller of Genome Analysis Toolkit (GATK, version 3.7). The candidate region was determined based on the SNP index.

### 2.4. Development of SSR Marker

The initial localization of the determinate inflorescence gene *Brdt1* and the location of its homologous gene in *B. napus* chromosomes helped determine the approximate range of *Brdt1* on the A10 chromosome. The sequence segment was downloaded from the BRAD (http://brassicadb.org/brad/) database, and SSR loci were detected using SSRHunter 1.3. Using the Primer 3 software (Premier Biosoft International, Palo Alto, CA, USA) to design the SSR primers. SSR amplification was performed following the method described by Lowe [31]. The co-dominance of SSR markers was detected using 6% polypropylene gel electrophoresis.

### 2.5. Mapping

The *Brdt1* gene was mapped using the $BC_3F_1$ populations (847 individuals) and $BC_4F_1$ populations (1267 individuals). The SSR markers and individual phenotypes were analyzed using the JoinMap 4/MapDraw program, resulting in the construction of a partial linkage map for the chromosome region containing the *Brdt1* gene.

### 2.6. Cloning and Sequence Analysis of the Candidate Gene

The *Brdt1* gene was amplified from the gDNA of parents 515 and 520. Primer 3.0 software was used to design the specific primers (Bra.TFL1-orf-F/Bra.TFL1-orf-R) of the gene (Table S1). The amplification process was as follows: pre-denaturation at 95 °C for 3 min, followed by 35 cycles of denaturation at 95 °C for 30 s, annealing at 57 °C for 30 s, extension at 72.0 °C for 60 s, and terminal extension at 72 °C for 10 min. The amplified sequences were cloned using the PMD19-T vector and *E. coli* DH5$\alpha$ methods. The positive clones were verified using M13-specific primers (Tsingke Biotech, Beijing, China). The gDNA sequences were analyzed using DNAMAN8.0.

### 2.7. Plasmid Construction and Plant Transformation

To assess the functionality of the candidate gene, a primer was designed based on the *Bra009508* (*TFL1* homologous) sequences obtained from NCBI (National Center for Biotechnology Information (https://www.ncbi.nlm.nih.gov/)). The primers were modified to include the *EcoRI* and *PstI* enzyme restriction sites, as well as 15 bp sequences from both ends of the pCAMIBA2300 vector. The amplification of the gDNA fragment, which consisted of the upstream, full-length gene, and downstream sequences, was performed using Phusion Hot Start High Fidelity DNA Polymerase (NEB, Ipswich, MA, USA) and the recombinant primers (CEBra.TFL1-F/CEBra.TFL1-R, Table S1). The indeterminate inflorescence line 515 served as the source of DNA for this amplification. To create the complementation plasmid *pBraA10. TFL1: BraA10. TFL1*, a genomic fragment was sequenced and digested with *EcoRI* and *PstI*. The digested fragment was then ligated into the pCAMIBA2300 vector using a One Step Cloning Kit (Vazyme, Nanjing, China). Finally, the confirmed recombinant plasmid was introduced into GV3101 (*Agrobacterium tumefaciens*).

The inflorescence impregnation method was implemented to transform *Arabidopsis thaliana* (L.) Heynh. Specifically, the fused construct was introduced into the *A. thaliana tfl1-2* mutant. PCR was conducted to verify and identify the positive transgenic plants (PT-Bra.TFL1-F/PT-Bra.TFL1-R, Table S1).

### 2.8. RNA Extraction and qRT-PCR

Total RNA was extracted from 520 and NIL-520 materials at different developmental stages, including the 2-leaf seedlings, budding stage, bolting stage, and root, stem, and leaf tissues during the bolting stage. The extraction was performed using a TaKaRa MiniBEST Universal RNA Extraction Kit (TaKaRa, Dalian, China) with three biological replicates. The extracted RNA samples were frozen in liquid nitrogen and stored at −80 °C. First-strand cDNA was synthesized using the PrimeScriptTM RT Reagent Kit (TaKaRa, Dalian, China) following the manufacturer's protocol. The primers (qRT-braA10-1F/qRT-braA10-1R, Table S1) for qRT-PCR were designed using Primer-BLAST in NCBI (Primer designing tool (https://www.ncbi.nlm.nih.gov/tools/primer-blast/, accessed on 22 January 2024)). The actin gene was selected as the reference gene for the relative quantification of the candidate gene (Actin-F/Actin-R, Table S1). qRT-PCR was conducted using a CFX Opus 96 instrument (Bio-Rad, Hercules, CA, USA). The reaction system consisted of 25 μL, including 2 μL of gene-specific primer (10 ng μL$^{-1}$) (Table S1), 2 μL of cDNA (50 ng μL$^{-1}$), 12.5 μL of TB Green Premix Ex Taq II (TliRNaseH Plus) (TaKaRa, Dalian, China), and 8.5 μL of sterile water. The PCR conditions were 95 °C for 30 s, followed by 45 cycles of 95 °C for 5 s and 60 °C for 30 s. The data were processed using the $2^{-\Delta\Delta Ct}$ method.

## 3. Results

### 3.1. Observations of the SAM Apex in B. rapa

The formation of the SAM was observed using a stereoscopic fluorescence microscope and paraffin section (Figure 2). The aim was to determine the point at which the apices of indeterminate inflorescence and determinate inflorescence exhibit morphological differences during the growth process. This observation showed that *B. rapa* exhibits normal inflorescence differentiation at the SAM of indeterminate and determinate inflorescences during the two-to-six-leaf stage (Figure 2a–c,e–g). However, at the eight-leaf stage, indeterminate inflorescences continued to undergo normal inflorescence differentiation (Figure 2d), while the SAM of determinate inflorescences exhibited variation and could not differentiate inflorescence tissue normally (Figure 2h). The results of paraffin sections also indicated that during the two-to-six-leaf stage, the indeterminate and determinate shoot apex meristem displayed the same shapes (Figure 2i–k,m–o). However, in the eight-leaf stage, the indeterminate normal process continued (Figure 2i), and the determinate apices had already exhibited variation and could not differentiate into normal inflorescence meristems (Figure 2p). These findings provide morphological evidence for the formation period of determinate inflorescences and offer insights for observing determinate inflorescences in *B. rapa*.

### 3.2. Genetic Analysis

This study investigated and analyzed the growth habits of the $F_2$ and $BC_1F_1$ populations. It was observed that all $F_1$ individuals exhibited a complete indeterminate phenotype, indicating the dominance of indeterminate growth over determinate growth. In 515 × 520 $F_2$ plants, 297 plants displayed indeterminate growth while 27 plants showed determinate growth, consistent with a segregation ratio of 15:1. The $BC_1$ plants exhibited a segregation ratio of approximately 3:1 (indeterminate-to-determinate = 168:50) (Table 1). These findings suggest that determinate growth habits are controlled by two independently inherited recessive genes, and determinate inflorescence genes were tentatively designated *Brdt1* and *Brdt2*.

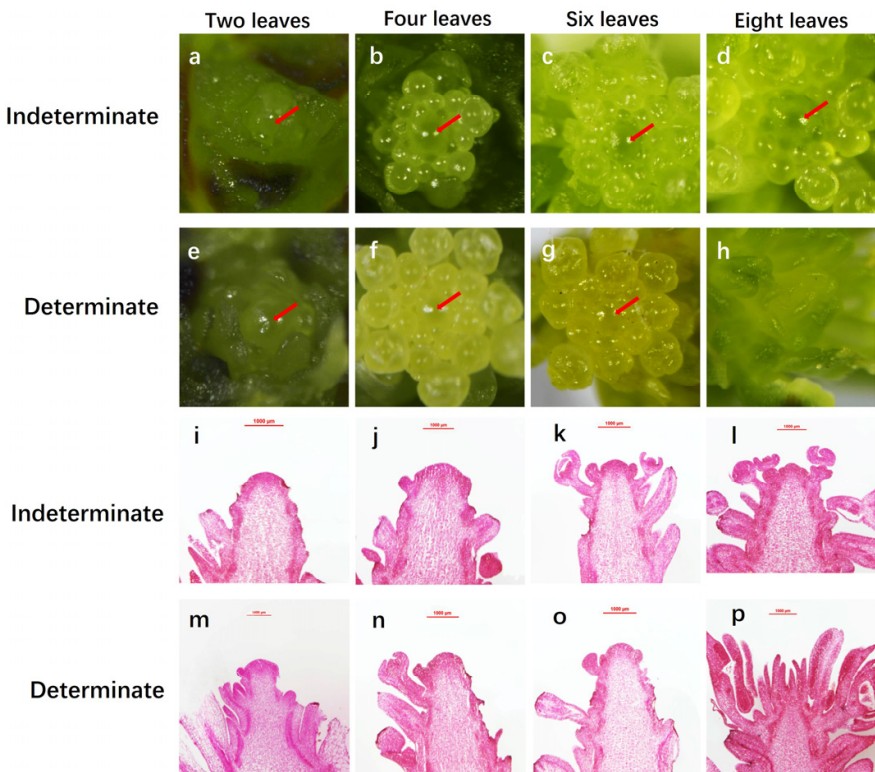

**Figure 2.** Microscope observation and paraffin section of the shoot apical meristem (SAM) of the indeterminate (**a**–**d**,**i**–**l**) and determinate inflorescences (**e**–**h**,**m**–**p**). (**a**,**e**,**i**,**m**) SAMs of indeterminate and determinate inflorescences at the two-leaf stage. (**b**,**f**,**j**,**n**) SAMs of indeterminate and determinate inflorescences at the four-leaf stage. (**c**,**g**,**k**,**o**) SAMs of indeterminate and determinate inflorescences at the six-leaf stage. (**d**,**h**,**i**,**p**) SAMs of indeterminate and determinate inflorescences at the eight-leaf stage. Red arrows indicate growth points. In the eight-leaf stage, the determinate inflorescence was beginning to appear.

**Table 1.** Segregation of inflorescence traits in the $F_2$ and $BC_1$ populations.

| Combination | F1/RF1 | Population | No. of INDT. Plants | No. of DT. Plants | Expected Ratio | $X^2$ Value |
|---|---|---|---|---|---|---|
| 515 × 520 | indeterminate | $F_2$ | 297 | 27 | 15:1 | 2.40 |
| | | $BC_1F_1$ | 168 | 50 | 3:1 | 0.49 |

### 3.3. Primary Mapping of the Determinate Gene Brdt1 Using BSA-Seq

The $BC_3F_1$ population obtained from a cross between 515 and 520 was used for BSA resequencing [32] to determine the position of the *Brdt1* gene. After sequencing the two bulks and their respective parents, a total of 54.429 Gb clean reads were obtained after quality control filtering. Specifically, the 515 parent, the 520 parent, the indeterminate bulk, and the determinate bulk accounted for 10.454 Gb, 10.731 Gb, 17.035 Gb, and 16.209 Gb, respectively. The sequencing data showed that more than 90.15% of the bases in both pools and parents had a quality score of more than 30 (Q30), and more than 96.13% had a quality score of more than 20 (Q20). Additionally, the average GC content was between 38.03% and 39.31%. The sequencing data were analyzed, and a total of 659,356 SNPs were identified between 515 and 520 by aligning with the 'chiifu' v1.5 reference genome. Out of these SNPs, 604,936 homozygous SNPs were found between the two parents, and these were used to calculate the SNP index for the two descendants. A graph of the ΔSNP index was plotted against the genomic regions (Figure 3a), revealing a significant peak in the 1.55 Mb region from 14.76 Mb to 16.31 Mb on chromosome A10 (Figure 3b). This finding suggests that the *Brdt1* gene, located in this specific region, could be a potential candidate locus.

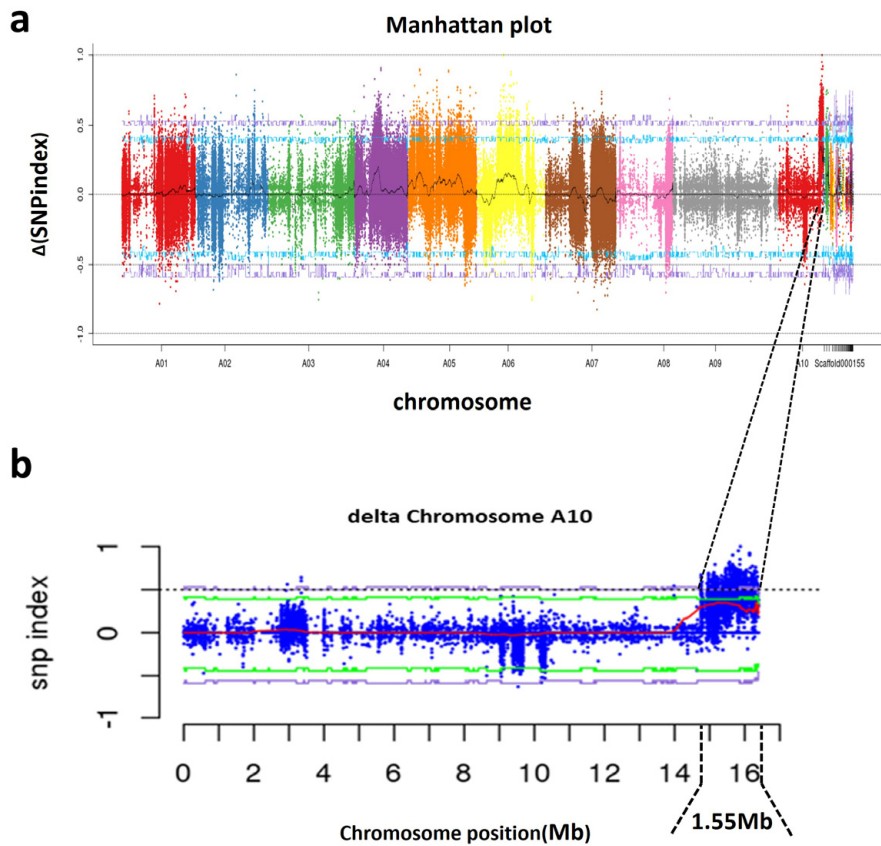

**Figure 3.** ΔSNP index Manhattan plot graphs. (**a**) The blue line indicates the 95% threshold value, and the purple line indicates the 99% threshold value. Different colors represent different chromosomes. (**b**) SNP index of delta chromosome A10. The red line is the SNP-index mean line, the green line is the 95% threshold line, and the purple line is the 99% threshold line.

### 3.4. Fine Mapping of the Brdt1 Gene

Based on the findings derived from the BSA-Seq analysis, it is likely that the *Brdt1* gene is located in the 14.76–16.31 Mb regions of A10 of *B. rapa*. Subsequently, the region sequence was downloaded from BRAD (http://brassicadb.org/brad/) and 60 SSR markers were developed. Among these, nine markers with polymorphism were identified and named BrSSR1 to BrSS9 (Table 2). A total of 847 BC$_3$F$_1$ individuals were screened using polymorphic SSR markers to assess the genetic distance between the *Brdt1* gene and the SSR markers and determine the order of these markers. Subsequently, the recombinants for each marker were recorded. The genetic distance was computed, and the genetic linkage map was constructed using JoinMap 4.0/MapDraw software V2.1. The results indicate that BrSSR1–BrSSR3 were located on one side of the *Brdt1* gene, BrSSR5–BrSSR9 were located on the other side of the *Brdt1* gene, and BrSSR4 co-segregated with the *Brdt1* gene (Figure 4a). Among the markers flanking the *Brdt1* gene, BrSSR3 and BrSSR5 displayed the closest linkage, with distances of 0.3 cM and 0.2 cM from the *Brdt1* gene, respectively (Table 2). These closely linked markers were subjected to BLAST analysis against BRAD (http://brassicadb.org/brad/). All markers were mapped to A10 of the 'chiifu' v1.5 reference genome (Table 2). Furthermore, the order of these markers on the map perfectly aligned with their counterparts on A10 of *B. rapa*. Based on this established order, the genomic region harboring the *Brdt1* gene was precisely delimited within an approximate interval of 232.3 kb, from 15,580.0 to 15,812.3 kb on A10 (Figure 4b).

**Table 2.** Information about the markers that were closely linked to *Brdt1*.

| Type of Marker | Name | Size of Marker | Physical Position (kb) | Chromosome of 'chiifu' v1.5 |
| --- | --- | --- | --- | --- |
| SSR | BrSSR1 | 174 | 15,476,191 | A10 |
| SSR | BrSSR2 | 158 | 15,505,412 | A10 |
| SSR | BrSSR3 | 185 | 15,580,043 | A10 |
| SSR | BrSSR4 | 166 | 15,712,914 | A10 |
| SSR | BrSSR5 | 175 | 15,812,313 | A10 |
| SSR | BrSSR6 | 215 | 15,859,041 | A10 |
| SSR | BrSSR7 | 182 | 15,872,725 | A10 |
| SSR | BrSSR8 | 238 | 15,912,798 | A10 |
| SSR | BrSSR9 | 234 | 15,918,222 | A10 |
| SSR | BrSSR10 | 174 | 15,731,259 | A10 |
| SSR | BrSSR11 | 151 | 15,785,574 | A10 |
| SSR | BrSSR12 | 182 | 15,793,126 | A10 |

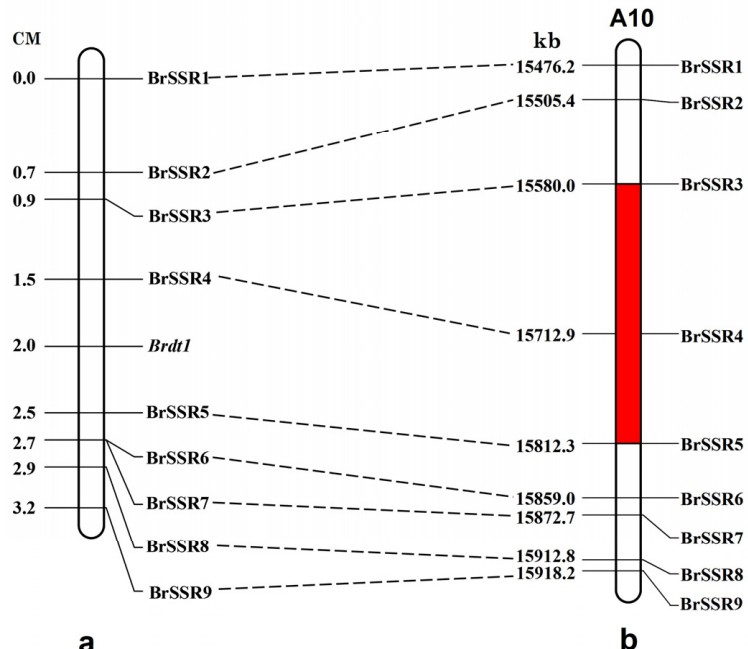

**Figure 4.** Mapping of *Brdt1* gene. (**a**) A partial genetic linkage map around the *Brdt1* gene. (**b**) A partial physical map of linkage markers around the *Brdt1* gene. The red region indicates candidate intervals, corresponding to 15,580.0–15,812.3 kb, identified through bulk segregant sequencing and SSR markers.

To narrow down the target region of *Brdt1*, a total of 1267 individuals from the $BC_4F_1$ population were used. Three SSR markers (BrSSR10–BrSSR12) were identified within the interval of 155,800 to 158,123 kb on A10 (Table 2). We utilized SSR markers including the previously used markers (BrSSR3–BrSSR5) and these SSR markers to conduct a screening of $BC_4F_1$ individuals. Our findings revealed that the *Brdt1* gene was positioned between BrSSR4 and BrSSR11 (Figure 5a). Using BLAST analysis against BRAD (http://brassicadb.org/brad/), the *Brdt1* gene was further narrowed down to an interval of approximately 72.7 kb, specifically between 15,712.9 kb and 15,785.6 kb on A10 of *B. rapa* (Figure 5b).

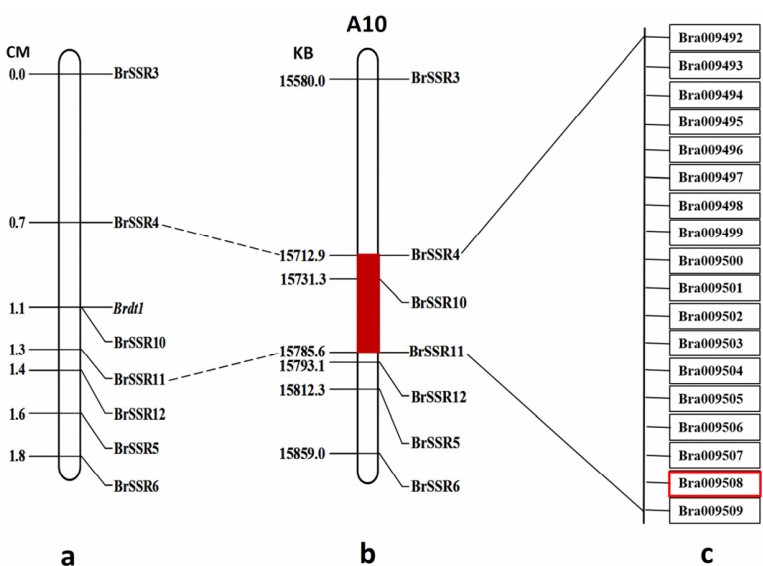

**Figure 5.** Fine mapping of *Brdt1* gene. (**a**) A partial genetic linkage map around the *Brdt1* gene. (**b**) A partial physical map of linkage markers around the *Brdt1* gene. The red region indicates candidate intervals corresponding to 15,712.9–15,785.6 kb. (**c**) Results of a BLAST analysis using sequences from candidate intervals against the 'chiifu' v1.5 genome.

### 3.5. Dissection of the Brdt1 Target Region

The candidate intervals were then submitted to the BRAD (brassicadb.cn) and TAIR (https://www.arabidopsis.org/) databases for BLAST analysis. The analysis revealed that the candidate region included 18 predicted genes (Figure 5c) from the *B. rapa* reference genome and showed homology to 17 genes on the *A. thaliana* chromosome (Table 3). Notably, according to the TAIR database of the gene annotation of these genes, the *Bra009508* (*BraA10.TFL1*) gene is homologous to the *AT5G03840* gene, which is known as the *TERMINAL FLOWER 1* (*TFL1*) gene. The *AT5G03840* gene encodes a phosphatidylethanolamine-binding protein (*PEBP*). In *Arabidopsis TFL1* mutants, determinate inflorescence can be formed at the SAM of the inflorescence. Based on this information, it was inferred that the *Bra009508* (*BraA10.TFL1*) gene was the most promising candidate gene for *Brdt1* and selected for further study.

### 3.6. Expression of BraA10.TFL1 in Different Tissues of B. rapa

A qRT-PCR analysis was conducted to investigate the expression levels of the *BraA10.TFL1* gene in different stages and tissues of *B. rapa* growth. Specifically, the gene was quantified in the shoot apex during the two-leaf stage, budding stage, and bolting stage of both indeterminate and determinate inflorescences. Additionally, gene expression was measured in the root, stem, and leaf tissues during the bolting stage. The findings revealed that the expression level of the *BraA10.TFL1* gene remained relatively consistent in the root, stem, and leaf tissues during the bolting stage. However, significant differences were observed in the expression of the *BraA10.TFL1* gene in the shoot apex between the 520 and NIL-520 lines, encompassing the two-leaf stage, budding stage, and bolting stage. Furthermore, the expression of *BraA10.TFL1* in the NIL-520 line was notably higher than in the 520 line, particularly during the P2 and P3 phases. These results emphasize the crucial role of the *BraA10.TFL1* gene in the development of distinct inflorescences (Figure 6). Therefore, it is reasonable to postulate that *BraA10.TFL1* is a potential candidate gene for the inflorescence trait.

**Table 3.** Results of BLASTN searches using the candidate interval gene.

| Gene of *B. rapa* | Homologous Gene in *A. thaliana* | Putative Function |
| --- | --- | --- |
| Bra009492 | AT5G04030 | unknown |
| Bra009493 | AT5G04020 | Calmodulin-binding |
| Bra009494 | AT5G04010 | F-box family protein |
| Bra009495 | AT5G03990 | FK506-binding-like protein |
| Bra009496 | AT5G03980 | SGNH hydrolase-type esterase superfamily protein |
| Bra009497 | AT5G00970 | F-box family protein |
| Bra009498 | AT5G03960 | IQ-domain 12 |
| Bra009499 | AT5G03940 | Chloroplast signal recognition particle 54 KDa subunit protein |
| Bra009500 | AT5G03910 | ABC2 homolog 12 |
| Bra009501 | AT5G03905 | Iron-sulfur cluster biosynthesis family protein |
| Bra009502 | AT5G03900 | Iron-sulfur cluster biosynthesis family protein |
| Bra009503 | AT5G03900 | Iron-sulfur cluster biosynthesis family protein |
| Bra009504 | AT5G03890 | unknown |
| Bra009505 | AT5G03880 | Thioredoxin family protein |
| Bra009506 | AT5G00893 | unknown |
| Bra009507 | AT5G03850 | Nucleic acid-binding, OB-fold-like protein s28 |
| Bra009508 | AT5G03840 | TFL1 (TERMINAL FLOWER 1); PEBP (phosphatidylethanolamine binding protein) family protein |
| Bra009509 | AT5G03795 | Exostosin family protein |

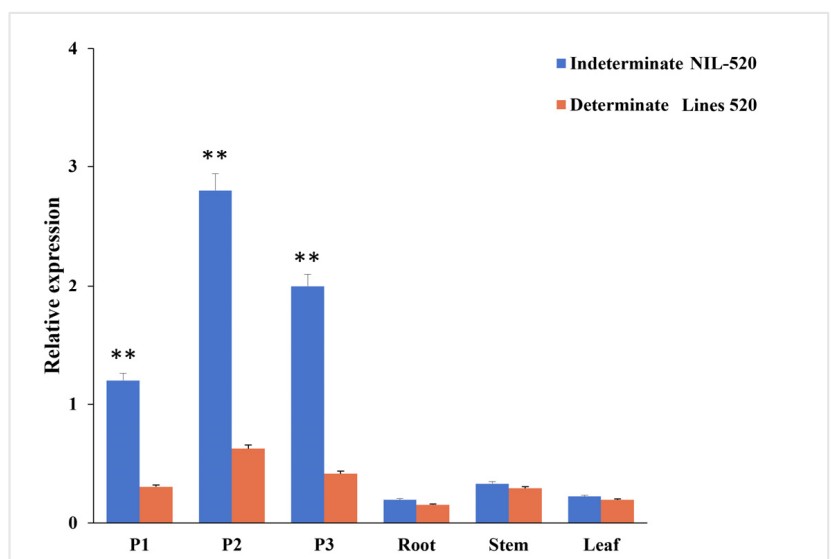

**Figure 6.** Expression analysis of the *BraA10.TFL1* gene among different stages and tissues of the NIL-520 and 520 lines in *B. rapa.* P1, P2, and P3 are two-leaf stage, budding stage, and bolting stage, respectively. The expression levels in the root, stem, and leaf tissues were consistent during the bolting stage. ** indicates significant differences at $p < 0.01$.

*3.7. Cloning and Sequencing Analysis of the BraA10.TFL1/BraA10.tfl1 Gene*

The sequences of gDNA and CDS of *BraA10.TFL1/BraA10.tfl1* were amplified from the indeterminate inflorescence line 515 and the determinate inflorescence line 520. DNA-MAN 8.0 software was used to analyze the gDNA and CDS sequences of the determinate and indeterminate inflorescences. The results revealed the presence of a gDNA sequence

measuring 1066 bp and a 537 bp cDNA sequence in inflorescence lines 515 and 520, respectively. These sequences contained four exons and three introns. The sequence analysis of *BraA10.TFL1* (BraA10.TFL1_DNA) and *BraA10.tfl1* (BraA10.tfl1_DNA) in *B. rapa* revealed two SNP mutations (G 434 T and C 569 T) in the intron region (Figure 7a). Amino acid sequence prediction and the analysis of the *BraA10.TFL1/BraA10.tfl1* genes were performed using Premier 5. The results showed no differences in amino acid sequences (Figure 7b).

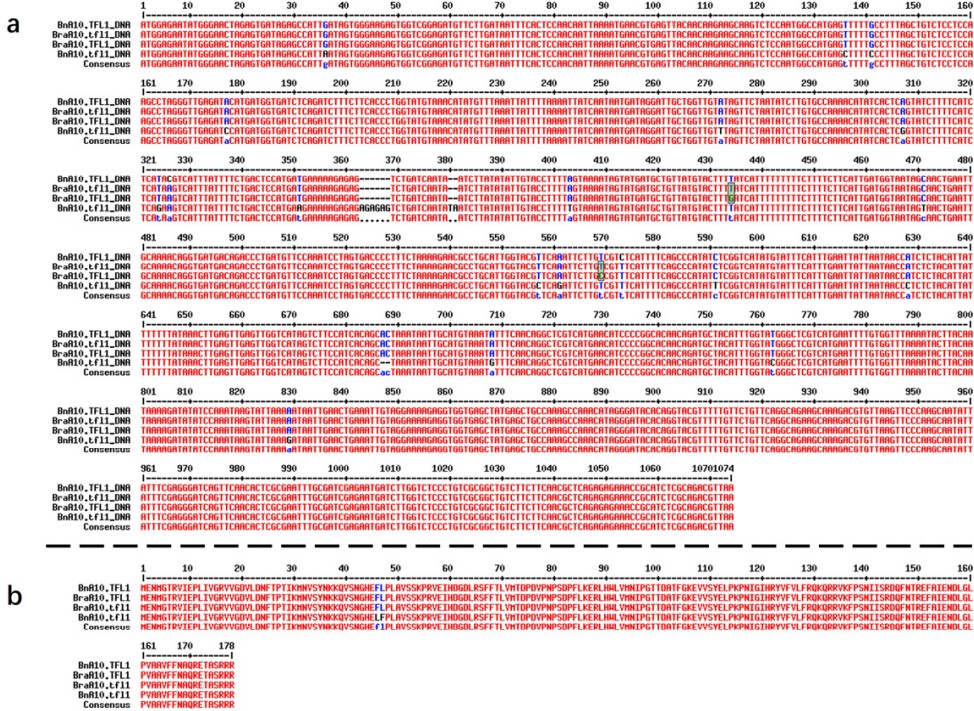

**Figure 7.** Sequence analysis of *BraA10.TFL1*. (**a**) The gDNA sequences of *BraA10.TFL1* (indeterminate) from the 515 line and *BraA10.tfl1* (determinate) from the 520 line were aligned with *BnA10.TFL1* (indeterminate) and *BnA10.tfl1* (indeterminate) from *B. napus*. The green box indicates the position of SNP differences between indeterminate 515 and determinate 520. Blue indicates the bases that are different in the sequence. (**b**) Amino acid sequence alignment between *BraA10.TFL1* (indeterminate) and *BraA10.tfl1* (determinate) with *BnA10.TFL1* (indeterminate) and *BnA10.tfl1* (indeterminate) from *B. napus*. Blue indicates amino acids with differences in the sequence.

Previous studies have reported that the gene *BnA10.tfl1*, responsible for determinate inflorescence, is located on the A10 chromosome of *B. napus*. A comparison of the sequences of the *BnA10.tfl1* and the *BnA10.TFL1* gene, which controls indeterminate inflorescence, revealed 22 differences [33]. Among these differences, two SNPs resulted in a change in two amino acids (Phe to Leu and Leu to Phe), potentially leading to a transition from indeterminate to determinate inflorescence. To further investigate this, the gene and amino acid sequences of *BraA10.TFL1* and *BraA10.tfl1* from *B. rapa* were compared with the *BnA10.TFL1* and *BnA10.tfl* genes from *B. napus*. The results indicated that *BraA10.tfl1* and *BnA10.TFL1* had the same size, consisting of 1066 bp, and showed high homology with only two differences out of 1066 bases. However, there were 20 sequence differences between *BraA10.tfl1* and *BnA10.tfl1*, resulting in changes in two amino acids (Phe to Leu and Leu to Phe). These unexpected findings highlight the importance of conducting functional validation to confirm the accuracy of the predicted genes.

### 3.8. BraA10.TFL1 Rescues the tfl1-2 Mutant Phenotype in A. thaliana

The complete genomic DNA sequence of *BraA10.TFL1* from the 515 line was amplified and utilized to construct *pBraA10.TFL1: BraA10.TFL1* in the PCAMBIA2300 vector. This vector contained a 3437 bp genomic fragment that included the *BraA10.TFL1* gene, with

1842 bp upstream, 1066 bp coding region, and 529 bp downstream (Figure S2). After the transformation process, we introduced this construct into the *A. thaliana tfl1-2* mutant (determinate inflorescence) (Figure 8b). Both the transgenic *A. thaliana* lines and *tfl1-2* mutants exhibited the expected outcomes (Figure 8b–d). The transgenic lines showed indeterminate inflorescence stem growth, similar to wild-type *A. thaliana* (Figure 8a). A total of 12 positive transgenic plants were obtained. The terminal flowers of *tfl1-2* mutants suppressed the differentiation of the SAM, resulting in the development of determinate inflorescence. It is important to note that the T1 *A. thaliana* plants displayed multiple buds, indicating an indeterminate phenotype (Figure 8c,d). In conclusion, these findings provide evidence supporting the functional similarities between *BraA10.TFL1* and *TFL1*.

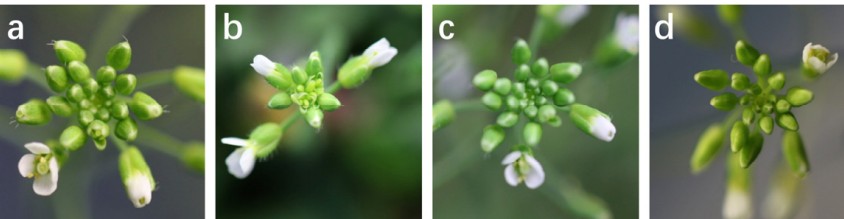

**Figure 8.** Architecture of terminal racemes in transgenic *A. thaliana*. (**a**) Wild-type *A. thaliana* (WT). (**b**) The *A. thaliana tfl1-2* mutant. (**c**,**d**) T1 *A. thaliana* transgenic plants returned to an indeterminate phenotype.

## 4. Discussion

Inflorescences significantly impact the yield of *B. rapa* [34–36]. In previous studies, the formation of determinate inflorescences changed plant height and the flowering and maturity stages. This not only improved the plant type, but also facilitated mechanized harvesting. Among the determinate mutant plants currently studied, *A. thaliana* [37], *B. juncea* [8], and *Vigna radiata* [38], it has been observed that the determinant inflorescence is regulated by a recessive gene. Zhang et al. [39] examined a natural mutant strain, FM8, of *B. napus* with a determinant inflorescence, and genetic analysis revealed that the inheritance of this inflorescence type is controlled by the interaction of two recessive genes and one recessive epistasis suppressor gene. Li et al. [14] found that the determinate inflorescence strain of *B. napus* was controlled by two independently inherited recessive genes (*Bnsdt1 and Bnsdt2*). However, the determinate inflorescence line 520 of *B. rapa* was used as material in this study, and it was found that it is controlled by two independent inherited recessive genes, *Brdt1* and *Brdt2*. This result may be due to the different mutant types of determinate inflorescence. Therefore, further in-depth research is necessary to understand the genetic mechanisms governing determinate inflorescence genes in rapeseeds.

This study utilized stereoscopic fluorescence microscopy and paraffin sectioning to observe the growth stages of the shoot apical meristem (SAM) in both determinate and indeterminate inflorescences. The results showed that the SAMs of determinate inflorescences (Figure 2e–g,m–o) were structurally similar to those of indeterminate inflorescences (Figure 2a–c,i–k) at the two-, four-, and six-leaf stages. However, the SAMs of determinate inflorescences (Figure 2h,p) showed differences compared to indeterminate inflorescences (Figure 2d,l) in the eight-leaf stage. The apex of the determinate SAM exhibited variation similar to the shape of a floral organ, which may cause determinate growth. Other studies have reported the observation of determinate SAMs forming terminal flowers, either singularly or in multiples [40,41]. While the phenotypes exhibited variation among different plants with determinate inflorescences, they had one fundamental characteristic in common: the shoot apical meristem (SAM) maintained its differentiation, producing fresh floral tissue, thereby causing a transition from an indeterminate to a determinate growth pattern.

In previous studies, the genes responsible for controlling quality traits were identified using bulk segregant sequencing (BSA-Seq) and map-based cloning approaches [42–46].

This study employed a method to map the determinate trait of *B. rapa*. Initially, the genomic sequence information of 'chiifu' v1.5 (*B. rapa*) was used as a reference to successfully map the *Brdt1* gene to the A10 chromosome. The *Brdt1* gene was delimited to an interval of approximately 72.7 kb, ranging from 15,712.9 kb to 15,785.6 kb on A10 of *B. rapa*. Furthermore, a gene annotation analysis within this interval identified a highly similar gene, *Bra009508* (*BraA10.TFL1*), which is homologous with the *TFL1* gene in *A. thaliana*. A sequence analysis of *BraA10.TFL1* revealed two SNP differences (G 434 T; C 569 T) in the intron region between indeterminate and determinate sequences. Subsequently, an expression pattern analysis of *BraA10.TFL1* was conducted, which demonstrated its specific expression in the shoot apex. Genetic transformation experiments in *A. thaliana* further confirmed the functionality of the *BraA10.TFL1* gene. When introduced into the *A. thaliana tfl1-2* mutant, the T1 *A. thaliana* plants reverted to an indeterminate state. Jia et al. [33] conducted a study in which they transferred the *TFL1* homologous gene *BnA10.TFL1* from *B. napus* into a determinate *Arabidopsis* mutant. The researchers observed that this transfer restored determinate expression in *Arabidopsis*, resulting in indeterminate inflorescence. These findings provide evidence that *BraA10.TFL1* shares a similar function to *TFL1*.

In this study, it is noteworthy that the sequence alignment results indicated no difference in the amino acid sequences encoded by the determinate inflorescence and indeterminate inflorescence sequences. However, the genetic transformation results demonstrated that the *BraA10.TFL1* gene from indeterminate inflorescence was successfully introduced into the *A. thaliana tfl1-2* mutant (regenerating plants from *B. rapa* through genetic transformation is challenging). As a result, the mutant's phenotype could be restored to indeterminate inflorescence. Based on these findings, it is hypothesized that the promoter region of the gene may contain a functional region that contributes to determinate inflorescence formation. However, additional evidence is necessary to substantiate this hypothesis.

## 5. Conclusions

As a novel combination of materials was used in this study, genetic analysis revealed that the determinate inflorescence is controlled by two independent recessive genes (*Brdt1* and *Brdt2*). Morphological observations indicated that the 520 strain with determinate inflorescence exhibited the characteristic at the eight-leaf stage. One of the genes, referred to as *Brdt1*, was mapped using BSA sequencing and SSR marker development. The gene was fine-mapped to the 15,712.9–15,785.6 kb interval on chromosome A10. Within this interval, it was found that the gene *Bra009508* (*BraA10.TFL1*) is homologous to the *AT5G03840* gene in *Arabidopsis*, which is annotated as the *TFL1* gene. Moreover, when the *BraA10.TFL1* gene from an indeterminate inflorescence plant was transferred to the *A. thaliana tfl1-2* mutant, the determinate traits became indeterminate. These findings suggest that *BraA10.TFL1* may play a role in controlling the determinate inflorescence trait. Overall, this research provides novel insights into the molecular mechanism of oilseed breeding for determinate inflorescences.

**Supplementary Materials:** The following supporting information can be downloaded at: https://www.mdpi.com/article/10.3390/agronomy14020281/s1, Figure S1: Population construction; Figure S2: The 3437 bp genomic fragment incorporating the *BraA10.TLF1* gene. Table S1: The primer sequences used in this study.

**Author Contributions:** C.C. and X.Z. performed the research and wrote the manuscript; Z.Z. conducted the data analysis. K.L. and D.D. designed the research and revised the article. All authors have read and agreed to the published version of the manuscript.

**Funding:** This research was financially supported by the Qinghai Provincial Natural Science Foundation of China (2022-ZJ-975Q).

**Data Availability Statement:** No new data were created.

**Acknowledgments:** We are grateful to Yongping Jia, Xutao Zhao, Lingxiong Zan, Liren Xie, and the Oil Crop Research Institute of the Chinese Academy of Agricultural Sciences for their help in purchasing *A. thaliana* mutants.

**Conflicts of Interest:** The authors declare no conflicts of interest.

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
