# Peer review of "Fine Mapping and Functional Verification of the Brdt1 Gene Controlling Determinate Inflorescence in Brassica rapa L."

_agronomy, doi:10.3390/agronomy14020281_

Round 1

Reviewer 1 Report

Comments and Suggestions for Authors

Line 64: Write terminal flower 1 in full in this line, and delete it from line 67

Line 67: Arabidopsis[18]. Its interaction ....substitute comma by dot

Line 70:  when highly expressed in Arapdopsis?

Line 89: Lines 515 and 520 of B. rapa from a breeding program? Germplasm bank?

Line 103: as containing the dominant locus BrDT1 (is this one among the 9/12?) ? Is this PCR? Sequencing?

Figure 1: 520 or 521?

Line 127: is it the locus and not an allele that should be present? Why ?

Line 129: When you say BSA here, is it by sequencing the bulks? Which is the sequencing methodology?

Line 238: two loci

Line 247: Is G for Gibgabases (Gb?)

Line 264: all SSR markers  within this region?

Line 265: found? Confirmed two be polymorphic?

Liine 281: what does constructed mean? Obtained individuals by crossing?

Line 301: e-value?

Fig 6. Beautiful. Amazing.

Line 334: CDS is for? Reverse transcription?

Line 334: BnA10.tfl1, from B. napus

Line 345: A comparison of the sequence of the BnA10.TFL1 gene with? Reference genome? Between Bn and Bra?

Line 392: Nicotiana benthamina ?

428 to 486 : the paragraph is an abstract, not discussion, but is okay

453 Maybe the intron is what make the difference, even if through promoter interaction? Could you review about the intron significance possibility?

Comments on the Quality of English Language

It is fine

Author Response

Dear Reviewer, We sincerely appreciate your valuable feedback on our manuscript. We have carefully considered your comments and suggestions and have made the necessary revisions accordingly. We have also provided detailed responses to each of your feedback points, which can be found in the uploaded PDF format. We are confident that these revisions have significantly enhanced the quality and clarity of our manuscript. Once again, we would like to express our gratitude for your insightful input. Best regards.

Reviewer 2 Report

Comments and Suggestions for Authors

The article is interesting and very innovative but it is necessary to keep in mind that the articles are written in the third person and the first person is used a lot. There are also tables and figures that do not appear in the text and figures that are not mentioned, everything is indicated in the document. It is important to describe the statistical procedure used in Figure 6 in the materials and methods.

Comments on the Quality of English Language

Only the grammatical problem of the first person is presented.

Author Response

(The authors gave the same response as above.)

Reviewer 3 Report

Comments and Suggestions for Authors

Review of the manuscript entitled:

“Fine Mapping and functional verification of the Brdt1 gene controls determinate inflorescence in Brassica rapa”.

The submitted manuscript is focused on the analysis of genome region of interest, concerned of determinate of inflorescence of B. rapa.  The idea of the research and methodology gives the insights in the genetic region structure and developed genes of interests are validated and verified correctly. Generally authors precisely underlines the genetic position and function of the gene responsible for inflorescence traits and many aspects are provided in the research.

However, there are some inconsistencies and the manuscript may be accepted after some corrections.

In the Abstract use ‘localized’ instead of ‘situated’ –line 19

In my opinion the information from line 34 – to 37 are not necessary and authors could add short paragraph about the B. rapa genome and about the segregating population for mapping. See the review articles:  “Molecular Mechanism of Flowering Time Regulation in Brassica rapa: Similarities and Differences with Arabidopsis”, 2024, Na Li, Rui Yang, Shuxing Shen, Jianjun Zhao: https://doi.org/10.1016/j.hpj.2023.05.020

and Willeke Leijten, Ronald Koes, Ilja Roobeek,1 and Giovanna Frugis3, 2018. Translating Flowering Time from Arabidopsis thaliana to Brassicaceae and Asteraceae Crop Species;  https://doi.org/10.3390/plants7040111

In material preparation section It would be good to add the scheme of the plants cross-breeding combination, which genotypes were selected for each step of the analysis.

There was no supplementary data connected with the manuscript so I do not considered the figures S1, Tables S1 et cet.

In the description of figure 1 there is Line 521??? – Probably mistake or additional hybrid line was added in the research.

Line 115 material and methods it is not necessary to use the information of previous studies it should be in Discussion section. I would propose to change the sentence into: The inflorescence variations was observed under the fluorescent microscope base on stem apex meristem development in accordance to the method described by Kobayashi et al. [27]

In paragraph 2.5 in my opinion the Bin Mapping procedure was applied.

In table 2 there is a mistake in the BRSSR9 position on chromosome A10 15,918,222 on the physical map there is 15,928,100.

In the table 3 check the functional assignment of the genes, in my opinion there are some incorrect annotations.

In the paragraph 3.8 I understand the authors are focused on the knockout of the tfl1 gene in Arabidopsis thaliana transformed with the construct with the gen of interest –this is to confirm their functions and see the determinate of inflorescence.

However, I do not understand the paragraph on transformation of Nicotiana benthamiana leaves to see the subcellular gene position (nucleus and plasma). What was the purpose of this part of the research? I summarized that the gene determine the inflorescence constitution. Nothing about this research is mentioned in the discussion. If author decide that this is important, must add some comments to this part.

In the diagram on figure 6 - I suggest to add the names of the Lines 520 and NIL-520 in the legend.   

Comments on the Quality of English Language

None

Author Response

(The authors gave the same response as above.)

Reviewer 4 Report

Comments and Suggestions for Authors

Article title (lines 2-4).

I recommend writing in the title the full specific name of the crop that the authors studied

Abstract (lines 12-29). In general, it corresponds to the material of the article. However, I have a wish.

I recommend that the authors strengthen two sections: Abstract and Conclusion. It is necessary to emphasize the originality and novelty of the results they obtained for the first time. They carried out original crossings, obtained unique forms, conducted a number of morphological and genetic studies for the first time, and obtained unique and reproducible results important for practical selection.

Keywords correspond to the content and results of the presented studies (lines 30-31).

However, I recommend expanding the list by adding an indication of the study of morphology using microscopy methods: stereoscopic fluorescence microscopy, methods of paraffin sections.

1. Introduction (lines 33 -86).

The literature review is quite complete and gives an idea of the direction of research on the genetics of crop inflorescences. Modern literary sources are presented quite well.

At the end of the literature review, the authors’ state the result obtained in the work (lines 83-85). It is better to formulate the purpose of the work here, and transfer the statement of results to the Conclusion.

2. Materials and methods (lines (87-205)

In general, the section contains fairly detailed information about working methods and has a logical structure. The techniques are reproducible. The research was carried out at a modern high-tech level.

2.1. Plant Material and Population Construction (88-113)

From the text of the section it is clear that experiments to obtain material for research were carried out over several years. However, years of research are missing from the text.

Note: For a clearer understanding of research methods, it is necessary to add years of research into crossbreeding, producing hybrids, and so on.

2.2. Morphological observation and Paraffin Sectioning (114-122)

It should be noted what magnification was used to obtain confocal microscopy images.

2.7. Plasmid Construction and Plant Transformation (157-174)/

Indicate the full name of the species at first mention: Arabidopsis thaliana (L.) Heynh (line 171).

Indicate what methods the authors used when conducting research, in particular when developing a primer, introducing a recombinant plasmid into GV3101 (Agrobacterium tumefaciens), inflorescence impregnation method, etc.

3. Results (lines 206-397)

In general, the section contains fairly detailed information about the results of the work and has a logical structure. The results obtained are presented in detail and convincingly. They contain new scientific data that are of undoubted interest to practicing breeders.

The authors presented the necessary amount of tabular material and figures regarding the research results. Tables and figures fully illustrate the results.

4. Discussion 398-455.

The section provides a fairly brief summary of the results obtained, their discussion and comparison with data obtained by other authors.

5. Conclusion 456-468

In general, the section summarizes the work done. However, it is written very briefly and does not give an idea of the importance, originality and scientific novelty of the research carried out. I recommend that the authors strengthen this section and show what is the originality and novelty of their results.

References (483-583)

The list of references corresponds to the content of the article, is quite complete, and does not contain excessive self-citation. Contains links to 22 literary sources out of 45 over the past 5 years.

Thus, I believe that the article has an undoubted scientific novelty, importance for practical selection and genetics of the Brassica rapa culture.

Author Response

(The authors gave the same response as above.)
